# Effects of Added Dietary Fiber and Rearing System on the Gut Microbial Diversity and Gut Health of Chickens

**DOI:** 10.3390/ani10010107

**Published:** 2020-01-08

**Authors:** Linyue Hou, Baosheng Sun, Yu Yang

**Affiliations:** Laboratory of Animal Production, College of Animal Science and Veterinary Medicine, Shanxi Agricultural University, Taigu 030801, China; xinxiaobao85@163.com (L.H.); sxauLynn@163.com (B.S.)

**Keywords:** dietary fiber, lignocellulose, rearing system, free-range chickens, microbial diversity, gut microbiota, gut health, SCFAs, mucus layer, welfare

## Abstract

**Simple Summary:**

In recent years, more and more research has focused on the effects of free-range rearing on the welfare of chickens. However, few works have focused on the gut microbial diversity and gut health of free-range chickens, especially when plant fibers are lacking in the wild. A lack of dietary fiber decreases the gut microbial diversity and even damages the gut health of a host, so we added eubiotic lignocellulose to the feed of caged and free-range Chinese local Bian chickens at three different levels (0%, 2%, and 4%) from September to November, aiming to observe the effects of added dietary fiber and the rearing system on the gut microbial diversity, microbial metabolism, and gut health of chickens, as well as to determine an appropriate amount of lignocellulose to effectively improve the gut microbial diversity and gut health of chickens when available plant fibers are scarce. The results showed that adding 2% lignocellulose was appropriate for free-range chickens, while 4% lignocellulose was appropriate for caged chickens. In addition, compared with the 2% lignocellulose in the cage system, 2% lignocellulose in free-range rearing could effectively improve unique microbial diversity and gut development. Adding an appropriate amount of dietary fiber may be good for the gut microbial diversity and gut health of caged chickens and free-range chickens who suffer from a lack of plant fibers in the wild.

**Abstract:**

It is of merit to study the appropriate amount of dietary fiber to add to free-range chickens’ feed to improve their microbial diversity and gut health in times of plant fiber deprivation. Lignocellulose is a useful source of dietary fiber, and its positive effects on the growth performance and laying performance of chickens has already been proven. However, few researchers have researched the effects of adding it on the gut microbiota of chickens. In this research, we added three different levels of eubiotic lignocellulose (0%, 2%, and 4%) to the feed of caged and free-range Bian chickens from September to November, aiming to observe the effects of added dietary fiber and different rearing systems on the gut microbial diversity and gut health of chickens, as well as to determine an appropriate amount of lignocellulose. The results showed that adding dietary fiber increased the thickness of the cecum mucus layer and the abundance of *Akkermansia* and *Faecalibacterium* in caged chickens, and 4% lignocellulose was appropriate. In addition, adding lignocellulose increased the microbial diversity and the abundance of the butyrate-producing bacteria *Faecalibacterium* and *Roseburia* in fee-range chickens. The α-diversity and the length of the small intestine with 2% lignocellulose in free-range chickens were better than with 2% lignocellulose in caged chickens. Maybe it is necessary to add dietary fiber to the feed of free-range chickens when plant fibers are lacking, and 2% lignocellulose was found to be appropriate in this experiment. In addition, compared with caged chickens, the free-range chickens had a longer small intestine and a lower glucagon like peptide-1 (GLP-1) level. The significant difference of GLP-1 levels was mainly driven by energy rather than short chain fatty acids (SCFAs). There was no interaction between added dietary fiber and the rearing system on SCFAs, cecum inner mucus layer, and GLP-1.

## 1. Introduction

In recent years, animal welfare has attracted more and more social attention. The “World Farm Animal Welfare Conference—Beijing consensus”, which was proposed by 16 organizations including the Food and Agriculture Organization of the United Nations (FAO), was officially released in Beijing in 2018, and the third World Conference on Farm Animal Welfare (WCFAW) was held in Qingdao in 2019. In terms of chicken welfare, the European Union had completely abolished the traditional cage mode of laying hens by 2012. Free-range chickens can access natural environments and more space, and they have more opportunities to show natural behaviors such as foraging and sand bathing [1]. Free-range rearing is not only beneficial to welfare [2,3], it is also beneficial to the gut microbial diversity of chickens. Compared with caged chickens, free-range chickens who forage various plants have more abundant gut microbiota [4,5] and more Actinobacteria [6]. Varied and healthy gut microbiota depend on the diversity of the diet [7].

However, the decrease of food source diversity and dietary fiber content can reduce the gut microbial diversity of the host [8,9]. The gut microbiota of mice who were fed low dietary fiber long-term lost diversity, and this was seriously compounded over generations [10]. The loss of gut microbiota results in some diseases. For example, both obesity and type 2 diabetes have a common feature, which is the reduction of gut microbial diversity [11,12,13]. Therefore, gut microbial diversity has been a major focus of the Human Microbiome Project (HMP) [14,15]. In contrast, people who consume more dietary fiber have more gut microbiota [16,17], because dietary fiber is the substrate of bacteria and can be degraded into glucose and other monosaccharides, which provide carbon and energy sources for the proliferation of other microorganisms [18]. Some of these monosaccharides can be fermented to form short chain fatty acids (SCFAs) by microbiota, mainly including acetate, propionate and butyrate. Acetate is used as energy for peripheral tissues. Propionate produces glycogen in the liver, and butyrate provides ~70% energy for normal colonic epithelial cells [19] and promotes their proliferation [20]. Butyrate-producing bacteria include *Faecalibacterium*, *Roseburia*, *Coprococcus*, and *Anaerostipes*. *Faecalibacterium prausnitzii* [21] and *Roseburia intensis* [22] can use acetate to produce butyrate. Butyrate can increase the thickness of the mucus layer, preventing the invasion of pathogenic bacteria [23] and maintaining gut health. In addition, SCFAs can activate intestinal gluconeogenesis (IGN) [24,25] and induce the release of glucagon like peptide-1 (GLP-1) to make the host feel satiety and reduce food intake [26,27]. Therefore, suitable dietary fiber can improve the gut microbial diversity, the production of SCFAs, and the gut health of the host.

The Bian chicken is a Chinese native breed that is suitable for free-ranging. However, the available plant fiber reduces over time while chickens are in free-ranging systems, especially from autumn to winter, and a lack of plant fiber decreases the microbial diversity and gut health of the chickens. Therefore, we hypothesized that it is necessary to add dietary fiber to the feed of free-range chickens when they suffer from plant fiber deprivation. However, doing so is not completely safe, as some kinds of dietary fiber, such as the soluble dietary fiber inulin (though not insoluble fiber cellulose) has been shown to induce cancer in mice with disordered gut microbiota [28]. This highlights the importance of choosing the right kind of dietary fiber, which, in this case, was eubiotic lignocellulose for poultry [29]. Especially, no great adjustment was required in terms of the composition of the feed due to its high fiber content, so generally adding 1.0–1.5% can positively affect the growth performance and laying performance of poultry [30]. In addition, it is a synergistic combination of soluble and insoluble fiber, so it can produce more lactate and butyrate than traditional non-fermentable fiber in vitro. However, there are few reports on its effects on the gut microbiota of chickens. Given this, we added three different levels of eubiotic lignocellulose to the feed of caged and free-range Bian chickens, aiming to research the effects of adding dietary fiber and different rearing systems on the gut microbial diversity, SCFAs, GLP-1, cecum mucus layer and gut health of chickens, to determine whether it was necessary to add dietary fiber to the feed of free-range chickens when plant fibers were lacking, and to determine an appropriate amount of fiber added in the feed. In addition, we aimed to observe whether there was an interaction between adding dietary fiber and rearing systems or not, as well as the main effects of the two factors on the above indexes.

## 2. Materials and Methods

### 2.1. Animals, Feed and Experimental Design

This research was approved by the Shanxi Agricultural University Animal Experiment Ethics Committee, and the license number was SXAU-EAW-2017-002Chi.001. A 2 × 3 experimental design was used in this experiment. A total of 108 1-day-old Bian male chickens with a 40 g average weight were chosen, and they were reared in an animal breeding house of Shanxi Agricultural University in June. They were randomly divided into three groups; each group had six replicates, and one cage was a replicate with six chickens per cage. In this experiment, OptiCell (OC) lignocellulose was chosen as a kind of dietary fiber to be added to the fed of chickens. Given that adding 1.0–1.5% of it can positively affect the performance of chickens [30], group one and group two were given 1% and 2%, respectively, on the basis of complete formula granulated feed (Jinzhong Shiyang Feed Ltd., Taigu, China) (Table 1). In contrast, the control group was not given it. The product was supported by the Beijing e-feed & e-vet cooperation. It was developed by Agromed Ltd. (Austria), and it is a type of eubiotic lignocellulose that is made from special fresh timber, such as sawing wood, wet wood shavings, and tree bark (moisture ~50%). The water content and parasite content of the fresh wood were tested before the subsequent processing. Next, qualified raw materials were dried until their moisture content was close to 9%. The dried materials were milled to an ultra-fine particle size with Westerkamp Sw900 and then screened with a Square Plansifer MPAP. The milled product was granulated at 80 °C, and then the granulation was broken and screened before weighing and packaging. The particle size of the screened samples was tested every 4 h to ensure that the particle size met the requirements. The type of product used in this experiment was OC C5 with 25 kg per bag. Approximately 0%, 60–63%, 32–34% and 5–7% of particle sizes passed through >2000 mμ, >1000 mμ, >500 mμ and <500 mμ sieves, respectively. The precise ingredients of the product are shown in Table 2.

At the beginning of the ninth week, the chickens were regrouped. Some of chickens were transferred into three-step cages to continue being caged with the same group as before. These chickens were divided into three groups, with six replicates per group; one cage was regarded as a replicate, with three chickens per cage. Some of the chickens were free-range from September to November. A grassland was divided into three equal areas with net and wood stumps, and three chicken nests were built before the free-range Bian roosters (FBR) came in. Each area was a group, and each group contained 12 chickens. The plant fiber gradually reduced in the wild environment during this period.

We mixed different amounts of lignocellulose (0%, 2%, and 4%) (Beijing), corn, soybean meal, bran and premix (Taigu, Shanxi) to make three kinds of feed with equal energy and protein but different amounts of fiber (Table 2). Both of the rearing systems involved three kinds of feed during weeks 9–20 (Table 2). Group one had 2% added lignocellulose and was called the lignocellulose-low (OL) group, and Group two had 4% added lignocellulose and was called the lignocellulose-high (OH) group. The control group was the lignocellulose-free (OF) group. For simplicity, “caged Bian roasts-twenty weeks” was named CBRT for short, and “free-range Bian roasts-twenty weeks” was named FBRT. In short, CBRT contained the CBRT-OL, CBRT-OH and CBRT-OF groups. FBRT contained the FBRT-OL, FBRT-OH and FBRT-OF groups. This experiment was completed at the end of 20 weeks. Samples were collected to measure the microbial diversity, the thickness of the cecum mucus layer, and the levels of SCFAs and GLP-1.

### 2.2. Management

The chickens were caged in brood cages for the first 1–8 weeks and given free access to water and feed. The management of the indoor temperature, light, and humidity was conducted according to the breeding manual of chickens. No conventional immunization schedule of chickens was performed to avoid impacts on microorganisms. From week 9 to week 20, the caged chickens were fed twice a day, morning and night, with free water and food access. The chicken manure was cleaned in a timely manner. The free-range chickens were also fed twice a day. However, they were only fed 60% of the daily feed in the morning to encourage them to eat more plant fiber, and they were fed before they returned to the nests in the evening. The free-range area was about 15 m^2^ per chicken.

### 2.3. Sampling

We chose six chickens per group at 20 weeks old to collect blood samples from the wing vein after the chickens were fasted for 12 h. Then, the blood tube vessels were bathed in water at 37 °C for 1 h, followed by 3000 × g for 12 min. Following this, the upper serum was absorbed with a pipette into 0.5 mL EP tubes before being immediately preserved at −20 °C until further GLP-1 analysis.

Following this, the chickens were executed via humanitarian slaughter, and the length and weights of the intestines were measured. We quickly sampled two pieces from the middle of the right cecum and put them into a Carnoy’s fixative solution (the ratio of dry methanol:chloroform:glacial acetic acid was 60:30:10) for 5 h, and then we changed the samples into fresh Carnoy’s solution for 3 h before washing them in anhydrous methanol for 2 h. Finally, they were changed into new anhydrous methanol and stored at 4 °C until the thickness of the mucus layer was measured [31]. The right cecum contents and hypothalamus were collected into different cryogenic tubes, they were put into a liquid nitrogen tank, and then they were preserved at −80 °C until the determination of SCFAs. As above, the left cecum contents were collected to perform the 16S rRNA gene sequence of gut microbiota.

### 2.4. Determination

#### 2.4.1. 16S rRNA Gene Sequence

The 16S rRNA gene of gut microbiota was sequenced by Genedenovo Biotechnology Ltd. (Guangzhou, China) by using high-throughput sequencing technology. First, microbial DNA was extracted by using the HiPure Stool DNA Kits (Magen, Guangzhou, China). The V3–V4 regions of the 16S ribosomal RNA gene were amplified by PCR by using primers 341F 5′-CCTACGGGNGGCWGCAG and 806R 3′-GGACTACHVGGGTATCTAAT [32]. The first PCR reactions were performed in triplicate with 50 μL of a mixture containing 5 μL of 10 × KOD buffer, 5 μL of 2 mM dNTPs, 1.5 μL of each primer (10 μM), 1 μL of KOD polymerase, and 100 ng of template DNA. The amplification procedure was as follows: 94 °C for 2 min, followed by 30 cycles at 98 °C for 10 s, 62–68 °C for 30 s, 68 °C for 30 s, and a final extension at 68 °C for 5 min. The second round of amplification was a 50 μL mixture containing 5 μL of 10 × KOD Buffer, 5 μL of 2 mM dNTPs, 3 µL of 25 mM MgSO4, 1 μL of joint primer (10 μM), 1 μL of KOD polymerase, and 100 ng of template DNA, up to 50 μL with ultra-pure water. The amplification procedure was 94 °C for 2 min, 98 °C for 10 s, 65 °C for 30 s, followed by 12 cycles at 68 °C for 30 s, and a final extension at 68 °C for 5 min.

The second step was quality control and reads assembly. Reads filtering, reads assembly, raw tag filtering, chimera checking and removal were successively conducted. Then, the effective tags were clustered into operational taxonomic units (OTUs) of  ≥97% similarity by using the UPARSE pipeline [33]. The tag sequence with the highest abundance was selected as the reprehensive sequence within each cluster. Between groups, a Venn analysis was performed in an r-project (version 3.4.1) to identify unique and common OTUs. Then, the taxonomy classification, and α-diversity and β-diversity analyses, and functional prediction of these OTUs were successively conducted. The α-diversity reflects the microbial diversity within a single sample, including ACE, Chao1, Shannon, and Simpson. The values of ACE and Chao1 reflected the community richness. Shannon and Simpson reflected the community richness and community diversity. In contrast, the β-diversity was used to compare the microbial diversity between different samples.

#### 2.4.2. The Concentration of SCFAs

The SCFAs were measured with an internal standard method with high performance gas chromatography (HPGC). First, we prepared the deproteinated solution containing an internal standard crotonic acid. We accurately weighed metaphosphoric acid of 25 g and crotonic acid of 0.6464 g, and then we put them into a 100 mL volumetric flask. Then, we prepared 100 mL of a mixed standard stock solution as follows. Acetate, propionate, butyrate, isobutyrate, isovalerate and valerate standards (Sigma Ltd., St. Louis, MO, USA) were added at amounts of 60, 40, 20, 5, 5 and 5 μL (Table 3), respectively, to a 100 ml volumetric flask that was topped up to 100 mL with ultra-pure water and was then preserved at 4 °C. The volatile fatty acid standard solution was prepared as follows: 0.2 mL of deproteinized metaphosphate solution containing crotonic acid was added to three 1.5 mL centrifuge tubes, and 1 mL of mixed standard stock solution was added to this. The peak area of crotonic acid in standard solution was measured.

Sample preparation: ~0.5 g contents of the cecum was added to nine times the weight of ultra-pure water, homogenated, and centrifuged at 10,000 rpm for 10 min, and the supernatant was removed. Then, 1 mL of supernatant sample was placed into a 1.5 mL EP tube, and 0.2 mL of mixed solution of crotonic metaphosphate was added and reacted for 3 h. Centrifugation at 12,000 r for 5 min was undertaken. The supernatant was instantaneously injected into the chromatograph with a 10 μL microinjector, and the injection volume was 1.0 μL. Then, the conditions were set as follows. The injection temperature was set at 220 °C. The split was 5, and the split ratio was 6. The constant current was 0.8 mL/min. The initial temperature was set at 70 °C, and the detector temperature was 220 °C. Tail blowing was 40 mL/min, hydrogen was 35 mL/min, and air was 350 mL/min. The concentration of a certain acid (mmol/L) = (peak area of certain acid of sample × peak area of crotonic acid in standard solution × mol concentration of certain acid) ÷ (peak area of crotonic acid in sample × peak area of certain acid in standard solution).

#### 2.4.3. The Thickness of the Mucus Layer

Post Carnoy’s fixation, the methanol-stored right cecum samples were dealt with as follows. Washing, dehydration, transparence, embedding, slicing, deparaffinization and hydration to distilled water were undertaken. Then, sections were dyed with a AB-PAS Stain Kit according to the instructions. Pictures were taken for tissue slices with an Olympus imaging microscope. The thickness of the mucus layer was measured with IP WIN60 software or Image J software.

#### 2.4.4. GLP-1

We assayed the GLP-1 level in the serum by using a chicken GLP-1 ELISA kit (Solarbio Science & Technology Ltd., Beijing, China). The assay procedures are as follows. First, we prepared the original density standard according to the table provided by the instructions. Then, we began to add samples as follows. (1) Blank wells were set separately from testing sample wells. (2) A total of 50 μL of standard was added to microELISA strip plate, a sample dilution of 40 μL was added to the testing sample well, and then 10 μL of testing sample was added. (3) After closing the plate with a closure plate membrane, it was incubated for 30 min at 37 °C. (4) A 30-fold (or 20-fold) washed solution of configured liquid was diluted 30-fold (or 20-fold) with distilled water and reserved. (5) The closure plate membrane was uncovered, the liquid was discarded, and the plate was dried by swinging before adding washing buffer to every well. They were kept still for 30 s then drained, and this was repeated five times before drying by patting. (6) HRP-conjugate reagent of 50 μL was added to each well, except the blank well. (7) Step (3) was repeated. (8) Step (5) was repeated. (9) Chromogen solution A (50 μL) and chromogen solution B were added to each well, and they were protected from light for preservation for 10 min at 37 °C. (10) A stop solution (50 μL) was added to each well to stop the reaction (the reaction was stopped when the blue color changed to yellow). (11) Taking the blank well as zero, the absorbance was read at 450 nm within 15 min after adding the stop solution. (12) Taking the standard density as the horizontal and, the OD value for the vertical, the standard curve was drawn on graph paper. After finding out the corresponding density via the standard curve according to the sample’s OD values, it multiplied by the dilution multiple, and the result was the sample actual density. The unit was pmol/L.

### 2.5. Statistical Analysis

In terms of gut microbia, the α-diversity indexes, such as ACE, Chao1, Simpson, and Shannon, were calculated in QIIME. The comparison between groups was calculated by Welch's t-test and the Wilcoxon rank test by using the Vegan package (version 2.5.3) in R project, and the comparison among groups was computed by Tukey’s HSD test and the Kruskal–Wallis H test by using Vegan package (version 2.5.3) in R project [34]. The β-diversity analyses of Welch’s t-test, the Wilcoxon rank test, Tukey’s HSD test, and the Kruskal–Wallis H test were calculated by using Vegan package (version 2.5.3) in R project. The biomarker features in each group were screened by the linear discriminant analysis effect size (LEfSe) software (version 1.0) [35]. The functional prediction of the OTUs was inferred by using FUNGuild (version 1.0) [36].

Statistical analyses of SCFAs, the mucus layer, and GLP-1 were performed with a two-way analysis of variance (ANOVA) with Statistical Product and Service Solutions (SPSS) 22.0 (IBM). The results are expressed as the means and pooled standard error of the mean (SEM).

## 3. Results

### 3.1. The Determination of Gut Microbiota Diversity of CBRT

First, we compared the gut microbiota diversity among groups under the same cage rearing system or free-range rearing system, aiming to observe the effect of adding dietary fiber on gut microbiota.

#### 3.1.1. OTUs of CBRT

The OTUs of CBRT-OL (blue) was less than those of CBRT-OH (green) and CBRT-OF (purple). This means the microbiota of CBRT-OL was the lowest (Figure 1). The three groups shared the majority of OTUs.

#### 3.1.2. Gut Microbial Diversity Indexes of CBRT

The α-diversity indexes of CBRT-OL were all extremely significantly lower than the other two groups (*p* < 0.01, Table 4), and this was consistent with the Venn diagram. However, the β-diversity was not significantly different among the CBRT groups (*p* > 0.05, Table 4).

#### 3.1.3. The LEfSe of Significant Different Microbiota among Groups of CBRT

LEfSe was used for finding out the OTUs of the microbiota biomarkers between groups through linear discriminant analysis (LDA). A value of LDA of certain microbes of >2 represents that the difference was significant between groups (Figure 2).

We observed that the number of dominant bacteria of each group was almost equal (Figure 2).

Because lignocellulose is a type of fiber and can be degraded and fermented by microbiota to produce SCFAs, we focused on the significantly different fiber-degradation bacteria, SCFAs-producing bacteria (especially butyrate producer), mucus-eroding bacteria, and beneficial bacteria between groups. At the genus level, the relative abundance of *Bacteroides* (41.83%) in CBRT-OL was higher than in CBRT-OH (26.21%) and CBRT-OF (29.59%); however, the abundances of *Rikenellaceae_RC9_gut_group* (6.06%) and *Alloprevotella* (0.043%) were lower than those in CBRT-OH (13.58%) (0.70%) and CBRT-OF (14.17%) (0.71%) (Figure 2a,b). At the species level, the relative abundance of *B. sp_SB5* (25.12%) in CBRT-OL was higher than in CBRT-OH (6.26%) and CBRT-OF (5.99%); however, *B. gallinaceum* (0.19%) was lower than in CBRT-OH (3.85%) and CBRT-OF (7.95%) (Figure 2b,c). The results showed that adding dietary fiber did not consistently increase the relative abundance of the above potential fiber-degradation bacteria.

Compared with CBRT-OF, CBRT-OL and CBRT-OH both had dominant acetate-producing bacteria *Sutterella*, butyrate producer *Faecalibacterium* (Figure 2b,c) and beneficial bacteria *Akkermansia* (Figure 2b,c and Figure 3a,b), and CBRT-OH had more low abundance butyrate producers such as *Faecalibacterium*, *Coprococcus_1* and *Oscillospira* (Figure 2c). Notably, the model species *Akkermansia muciniphila* of *Akkermansia* is also a mucus-eroding microbiota.

### 3.2. The Determination of Gut Microbiota Diversity of FBRT

#### 3.2.1. OTUs of FBRT

The OTUs of FBRT-OL and FBRT-OH were both more than those of FBRT-OF. This means that adding dietary fiber could increase the gut microbial diversity of free-range chickens. The OTUs of FBRT-OL were the highest (Figure 4), and this means the gut microbial diversity of the 2% lignocellulose group was the highest.

#### 3.2.2. Gut Microbial Diversity Indexes of FBRT

The α-diversity of FBRT-OL was significantly higher than that of FBRT-OF, except for the Chao1 and Simpson indexes (*p* < 0.05, Table 5). The β-diversity of FBRT-OH was also significantly higher than that of FBRT-OF (*p* < 0.01, Table 5). This suggests that maybe adding dietary fiber could increase the microbial diversity of FBRT.

#### 3.2.3. The LEfSe of Significantly Different Microbiota among FBRT

Compared with FBRT-OF, FBRT-OL and FBRT-OH both had more of the low abundance butyrate producers *Roseburia* and *Faecalibacterium*. *Roseburia* can also effectively degrade dietary fiber (Figure 5a,b). Compared with FBRT-OF and FBRT-OH, FBRT-OL had a low abundance of dominant butyrate-producing bacteria *Megasphaera*. The number of butyrate producers in FBRT-OH was more than that in FBRT-OF, including *Faecalibacterium*, *Anaerostipes* and *Subdoligranulum* (Figure 5c). However, at the genus level, the relative abundance of *Bacteroides* (33.54%) in FBRT-OF was higher than that in FBRT-OL (19.55%) and FBRT-OH (20.07%) (Figure 5b,c). At the species level, the relative abundance of *B. caecigallinarum* (11.15%) in FBRT-OF was higher than that in FBRT-OL (2.69%) and FBRT-OH (2.71%) (Figure 5b,c).

#### 3.2.4. Comparison of Dominant Gut Microbiota among FBRT

FBRT-OL had more dominant bacteria, including the butyrate producers *Megasphaera* and fiber-degrader *Paraprevella*. FBRT-OH also had *Roseburia*, which is good at degrading fiber and producing butyrate (Figure 6).

### 3.3. Comparison of Gut Microbiota Diversity between CBRT and FBRT

In addition, we compared the gut microbiota diversity between caged chickens and free-range chickens at the same lignocellulose level, aiming to observe the effects of different rearing systems on them.

#### 3.3.1. Comparison of OTUs between CBRT and FBRT

The OTUs—especially unique OTUs—of the three groups of FBRT were more than those in CBRT, except for the FBRT-OF group (Figure 7). This indicated that only groups that had added dietary fiber increased their OTUs compared with CBRT, and it also indicated that it is necessary to add dietary fiber to the feed of free-range chickens, especially when plant fibers are lacking. However, we observed that, FBRT-OF had more unique bacteria genera and species than CBRT-OF. This was consistent with the knowledge that varied gut microbiota depend on the diversity of the diet such as plant fiber. Notably, the unique OTUs of FBRT-OL were more than that in CBRT-OL (Figure 7a).

#### 3.3.2. Comparison of Gut Microbial Diversity Indexes between CBRT and FBRT

The α-diversity indexes of FBRT-OL were all extremely significantly higher than those of CBRT-OL (*p* < 0.01, Table 6). However, the β-diversity showed no difference among the groups of CBRT (*p* > 0.05).

#### 3.3.3. The LEfSe of Significant Different Microbiota among CBRT and FBRT

The number of dominant bacteria of FBRT-OL and FBRT-OH were more than those in CBRT-OL and FBRT-OH, respectively (Figure 8a,b). The relative abundance of *Bacteroides* in CBRT-OL (41.83%) was higher than that in FBRT-OL (19.55%); however, the abundance of *Rikenellaceae_RC9_gut_group* (6.06%) was lower than that in FBRT-OL (13.16%) (Figure 8a). Compared with CBRT-OF, FBRT-OF had more of the dominant bacteria *Akkermansia* (Figure 8c and Figure 9).

Because the α-diversity in FBRT-OL and CBRT-OL was significantly different, we plotted the evolutionary branch tree diagram between them according to the LEfSe results (Figure 10). Compared with CBRT-OL (red), the dominant bacteria in FBRT-OL was Actinobacteria (green).

### 3.4. Functional Prediction of OTUs of CBRT-OL and FBRT-OL

Next, we performed a functional prediction of OTUs in CBRT-OL and FBRT-OL. This showed that the function of gut microbiota of CBRT-OL (Figure 11a) was similar to that of FBRT-OL (Figure 11b). Both were focused on carbohydrate metabolism, amino acid metabolism, the metabolism of cofactors and vitamins, energy metabolism, membrane transport, and signal transduction.

### 3.5. Comparison of the Development of Intestine between FBRT and CBRT

Rearing system but not added dietary fiber significantly affected the development of the intestine in this experiment. The length of the small intestine of FBRT was longer than that of CBRT by 9–17 cm, and that of FBRT-OL was significantly longer than that CBRT-OL (*p* < 0.05) (Table 7). The length and weight of the cecum of FBRT also was larger than that CBRT, except in the case of FBRT-OH. This suggests that free-range rearing benefited the development of the small intestines, while the development of the cecum was limited by high fiber.

### 3.6. Effects of Adding Dietary Fiber and Rearing System on SCFAs

There was no interaction between added dietary fiber and the rearing system, and the main effects of them on SCFAs were not significant (*p* > 0.05). There was no significant difference either in CBRT or FBRT, except that the isobutyrate of CBRT-OH was lower than that of CBRT-OF and FBRT-OH (*p* < 0.05) (Table 8).

In addition, we also detected the concentration of SCFAs in the hypothalamus. We found that acetate was the main SCFA. The average value of acetate in the three groups of CBRT was 1.27 ± 0.38, while in that of FBRT was 0.37 ± 0.028. Though the difference between them was not significant, the concentration in the hypothalamus in CBRT was higher.

### 3.7. The Thickness of the Cecum Inner Mucus Layer

Given the significant differences in the relative abundance of the mucus-eroding microbiota *Akkermansia*, we next observed the effects of adding dietary fiber and different rearing systems on the thickness of the cecum mucus layer. Only the inner mucus layer was measured in this study because the loose out mucus layer was lost. There was no interaction between added dietary fiber and rearing system. However, both the main effects of them were significant (Table 9).

#### 3.7.1. Effect of Added Dietary Fiber on the Thickness of the Cecum Inner Mucus Layer

The blue substance is the mucus layer and is indicated by the arrows in the figures (Figure 12). Compared with CBRT-OF, the cecum inner mucus layer in CBRT-OL (Figure 12a and Figure 13a) and CBRT-OH was thicker (Figure 13a), and the thickness of the cecum inner mucus layer in CBRT-OL was significantly greater than that in CBRT-OF (*p* < 0.05) (Table 9) (Figure 12 and Figure 13a). This suggests that adding dietary fiber could increase the thickness of the mucus layer of the cecum. This was consistent with the results that showed that the relative abundance of *Akkermansia* in CBRT-OL and CBRT-OH were both increased compared to that in CBRT-OF. There was no difference in FBRT in the thickness of the cecum mucus inner layer (Figure 13b).

#### 3.7.2. Effect of Rearing System on the Thickness of the Cecum Inner Mucus Layer

The thickness of the cecum mucus layer in CBRT-OH was greater than that in FBRT-OH (*p* < 0.05) (Table 9) (Figure 14 and Figure 15). This was consistent with the lesser development of the cecum in FBRT-OH.

### 3.8. GLP-1

Furthermore, we observed the effects of microbial metabolite SCFAs on GLP-1. There was no interaction between added dietary fiber and the rearing system on GLP-1 levels. There was no significant difference among any of the three groups of CBRT or FBRT (Table 10). The main effect of the rearing system on GLP-1 was significant, and the GLP-1 level of CBRT was higher than that of FBRT (*p* < 0.01) (Table 10). This suggests that caged chickens were more satiated compared to free-range chickens.

## 4. Discussion

### 4.1. The Effects of Added Dietary Fiber and Rearing System on Gut Microbial Diversity

Dietary fiber is a substrate of bacteria, and it provides carbon and energy sources for the proliferation of microorganisms [18]. Improving the fiber level can increase the gut microbial diversity of the host [16], while a big loss of gut microbiota can result in some diseases [11,12]. In this experiment, adding dietary fiber increased the gut microbial diversity of free-range chickens. In addition, the α-diversity of CBRT-OH was higher than that of CBRT-OL in the caged chickens, together suggesting that maybe adding dietary fiber could increase gut microbial diversity. However, the α-diversity of CBRT-OL was lower than that of CBRT-OF. This was related to the large-scale adjustment of bran in feed formula. Though the fiber content of CBRT-OF was the lowest, the content of bran was the highest. Bran is a fiber substrate of microbiota, so it increased the microbial abundance of CBRT-OF. This also suggests that we should not only focus on the fiber content while ignoring the effects of other components in feed formulas on microbial diversity.

The effect of rearing system on gut microbial diversity was also big. Research has shown that the gut microbial diversity of free-range chickens is higher than that of caged chickens [37]. This is mainly driven by consumption of various plant fibers. The diversity of herbivore bacteria is more abundant [38]. By contrast, a decrease in the diversity of food sources reduces the gut microbial diversity [8]. In this experiment, we also found that free-range chickens had more unique OTUs and gut microbiota than caged chickens, except for the OF group. This indicated that varied gut microbiota depends on the diversity of food, and it also highlighted that it is necessary to add dietary fiber to the feed of free-range chickens when plant fibers are lacking. In addition, we found that Actinobacteria, which is the main phylum in soil, was more abundant in the OL group of free-range chickens than in caged chickens, and this was consistent with a previous report [32]. Actinobacteria can produce various natural drugs and bioactive metabolites [32] that benefit the health of chickens. Free-range chickens usually look for worms in soil and forage gravels for grinding food, and so they can easily access Actinobacteria. In addition, the length of the small intestine of free-range chickens was longer than that of caged chickens, and this could be attributed to this special habit of eating gravel.

### 4.2. The Effects of Added Dietary Fiber and Rearing System on Akkermansia and the Cecum Mucus Layer

The mucus layer is the first physical barrier, and it can prevent aggression by enteric pathogens. The mucus layer is mainly composed of MUC2 mucin secreted by goblet cells [39]. The mucus layer consists of two layers; the outer mucus layer is loose and allows for microbial colonization, while the inner mucus layer is dense and contains little bacteria as a defense against enteric pathogens. The butyrate producer *Roseburia intestinalis* is specialized in colonization in mucin [40]. *Akkermansia muciniphila* is the model bacteria of *Akkermansia*. It is a typical mucus-eroding microbiota and also a beneficial bacteria. It has been found that increasing dietary fiber can increase the thickness of the mucus layer and the abundance of *A. muciniphila* [41], which protects the gut from colitis disease [42]. However, other research has shown that a lack of dietary fiber leads to an increase of *A. muciniphila* and the thinning of the mucus membrane [23].

In this experiment, there was no interaction between added dietary fiber and the rearing system; however, the main effect of added dietary fiber on mucus layer thickness was significant. In this experiment, the thickness of the cecum inner mucus layer and the relative abundance of mucus-eroding microbiota *Akkermansia* increased in both CBRT-OL and CBRT-OH. This indicated that added dietary fiber could increase these measures, and this was consistent with previous research [41]. This was because mucin is a potential growth substrate of *A. muciniphila* [43], the increase of the thickness of the mucus layer promoted *Akkermansia* reproduction, and these factors came together to improve the gut health of the host. Interestingly, *A. muciniphila* can also stimulate the production of mucus and the expansion of goblet cells that secret mucus [44], and it can increase the thickness of the mucus layer in turn. However, the causal association between the increase of the thickness of the mucus layer and the increased abundance of *Akkermansia* is still unclear for us. In addition, butyrate can also stimulate the synthesis of mucin glycoprotein [23]. Because the content of butyrate was not significantly different among groups in this experiment, the significant difference of the thickness of cecum inner mucus layer was not related to butyrate.

In addition, the main effect of the rearing system was also significant. The thickness of the cecum inner mucus layer in CBRT-OH was thicker than that in FBRT-OH, suggesting it may not be good for mucosal health of free-range chickens if too much dietary fiber is added. This was consistent with the lesser development of the cecum in FBRT-OH. This also suggests that the effects of adding dietary fiber may not be consistent in different rearing systems.

### 4.3. The Effects of Added Dietary Fiber and Rearing System on SCFAs

Dietary fiber can be fermented by microbiota to produce short chain fatty acids (SCFAs), including acetate, propionate, butyrate, isobutyrate, isovalerate, and valerate. Acetate, propionate, and butyrate account for 90%~95% of SCFAs. The acetate-producing bacteria *Bifidobacterium* produces acetate by using a bifid shunt [45]. Most propionate-producing bacteria such as *Propionibacterium* produces propionate via a succinate-propionate pathway [46]. However, butyrate-producing genera *Faecalibacterium*, *Eucharacterium* and *Roseburia* depend on butyryl-CoA:acetyl-CoA transferase to produce butyrate [47]. Eubiotic lignocellulose is a synergistic combination of soluble and insoluble fibers, so it can be fermented to produce more butyrate. Butyrate can maintain gut health. In this experiment, there was no interaction between the added fiber and the rearing system, and neither of two main effects had a significant effect on SCFAs. However, there was no significant difference in the concentration of acetate and butyrate among groups, though compared with the OF group, the dominant bacteria in the OL and OH group included the acetate-producing bacteria *Sutterella* and the butyrate-producing genera *Faecalibacterium* or *Roseburia*. This may be because the relative abundance of these dominant producers was low, or butyrate was absorbed more in the OL and OH groups.

### 4.4. The Effects of Added Dietary Fiber and Rearing System on GLP-1

In addition to supplying energy, SCFAs can also stimulate the release of glucagon like peptide-1 (GLP-1) and peptide YY (PYY) by activating GRP41 (renamed FFAR3) and GPR43 (FFAR2) to regulate appetite. GLP-1 is an incretin hormone that is secreted by intestinal endocrine L cells, making the host feel satiety and reducing intake.

There was no interaction between added dietary fiber and the rearing system. The main effect of the rearing system, rather than adding dietary fiber, on the GLP-1 level was significant. The GLP-1 level of the free-range chickens was lower in this experiment, indicating that they were hungrier than the caged chickens. Given that SCFA levels were not significantly different between CBRT and FBRT, we speculated that this was mainly caused by the energy of the diets rather than the SCFAs. The free-range chickens lacked energy, so they were hungrier. This showed that energy is also an important factor that affects GLP-1.

In addition, some studies have also proven that acetate could cause anorexia [48] through a central mechanism that is independent of GLP-1 via the hypothalamus. Therefore, we detected the content of acetate in the hypothalamus to test whether this difference between CBRT and FBRT in appetite was driven by this mechanism. However, we found that the value of acetate was not significantly different between CBRT and FBRT. Therefore, energy maybe the main factor that contributed to this difference in appetite, and this suggests that the energy of feed should be considered in experiments that discuss the effects of SCFAs on GLP-1.

## 5. Conclusions

Both adding dietary fiber and the chosen rearing system affect the gut microbial diversity and gut health of chickens. In this experiment, 4% lignocellulose was found to be appropriate for caged chickens, though it may not be beneficial for the gut health of free-range chickens, and 2% lignocellulose was found to appropriate for free-range when there is a lack of plant fibers in the wild. There was no interaction between added dietary fiber and the rearing system on SCFAs, the cecum inner mucus layer, or GLP-1 levels in this experiment; however, the main effect of different rearing systems on the development of the small intestine and GLP-1, especially fiber source diversity on gut microbial diversity, was greater than the added dietary fiber. Notably, the effects of other components besides crude fiber in feed formulas on microbial diversity should not be ignored, and the significant difference of GLP-1 levels was found to be mainly driven by energy rather than SCFAs in this experiment. In addition, further research is needed to determine whether there is the same trend when adding this product to the feed of different flocks of chickens.

## Figures and Tables

**Figure 1 animals-10-00107-f001:**
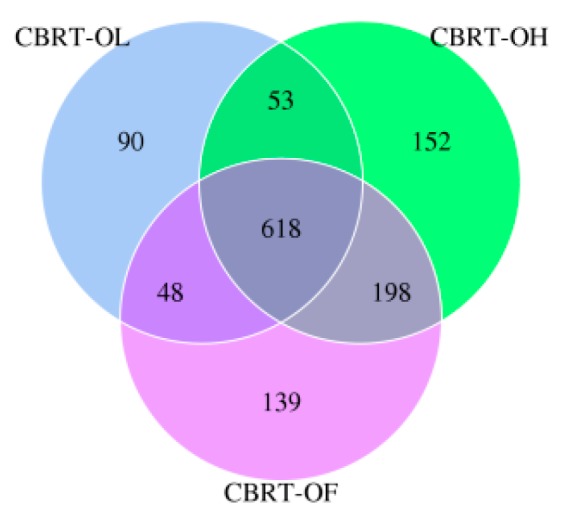
The Venn diagram of OTUs (operational taxonomic units) of CBRT (caged Bian roasts-twenty weeks).

**Figure 2 animals-10-00107-f002:**
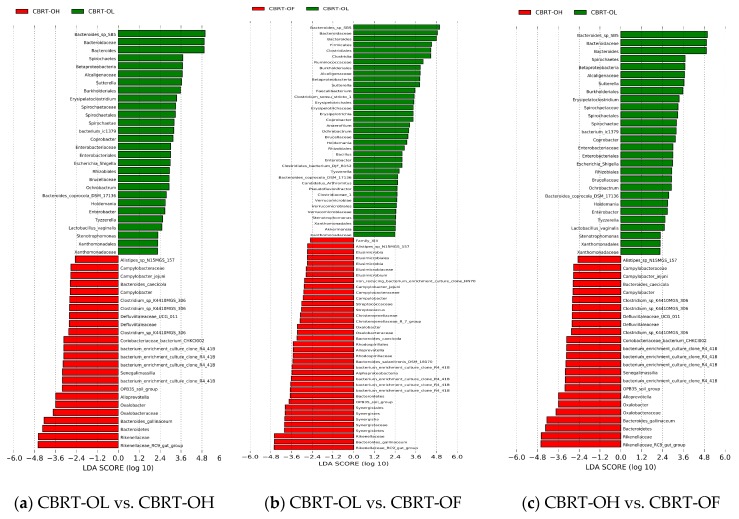
LDA (linear discriminant analysis) among the groups of CBRT.

**Figure 3 animals-10-00107-f003:**
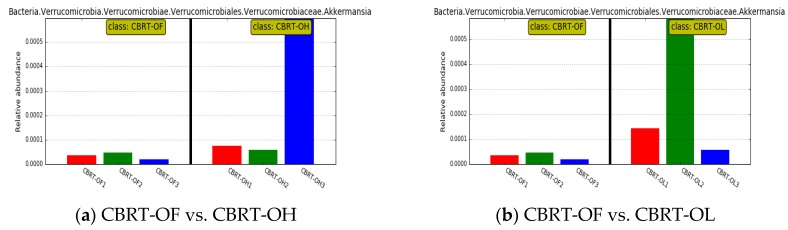
Abundance histogram of *Akkermansia* between the groups of CBRT.

**Figure 4 animals-10-00107-f004:**
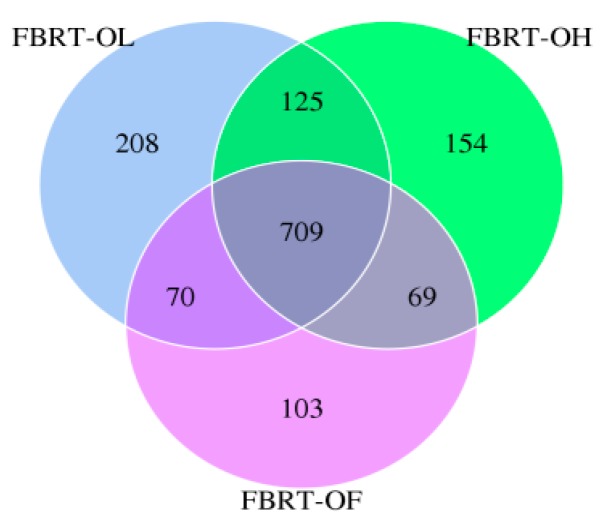
The Venn diagram of OTUs of FBRT (free-range Bian roasts-twenty weeks).

**Figure 5 animals-10-00107-f005:**
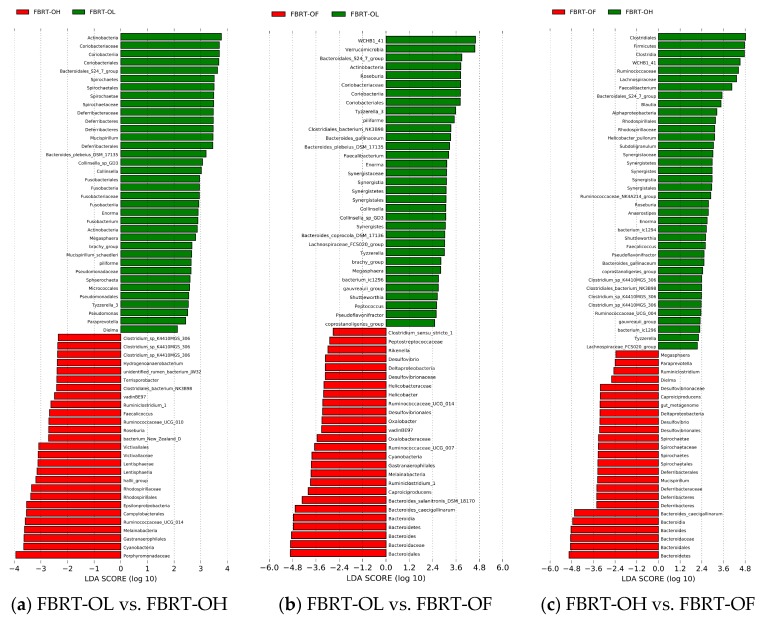
LDA among the groups of FBRT.

**Figure 6 animals-10-00107-f006:**
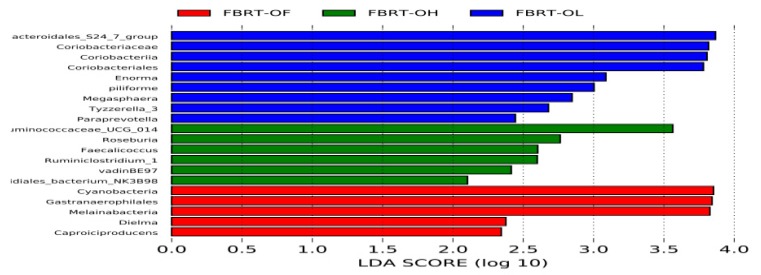
Abundance histograms of significantly different gut microbiota among FBRT.

**Figure 7 animals-10-00107-f007:**
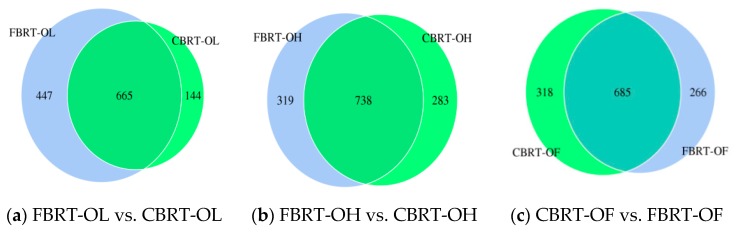
The Venn diagrams of OTUs of CBRT and FBRT.

**Figure 8 animals-10-00107-f008:**
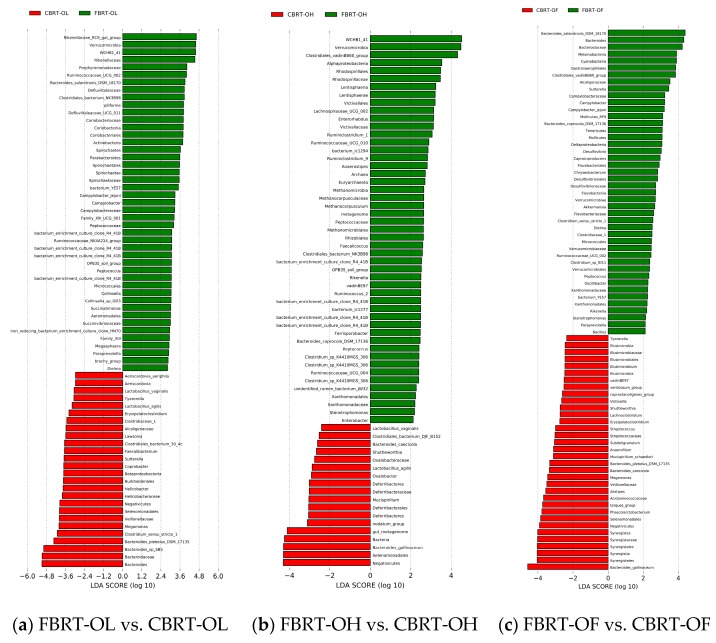
LEfSe (linear discriminant analysis effect size) of microbiota in FBRT and CBRT.

**Figure 9 animals-10-00107-f009:**
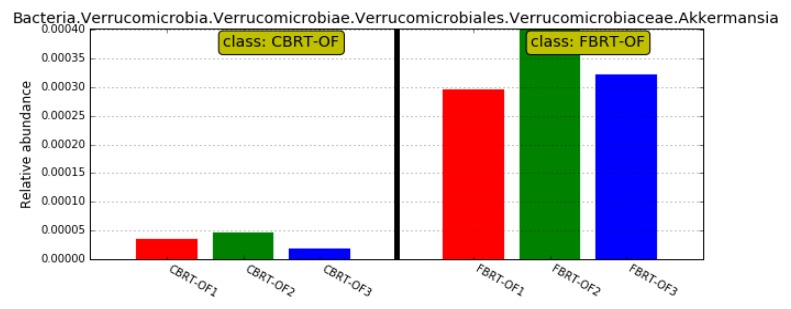
Abundance histogram of *Akkermansia* in CBRT-OF (lignocellulose-free) and FBRT-OF.

**Figure 10 animals-10-00107-f010:**
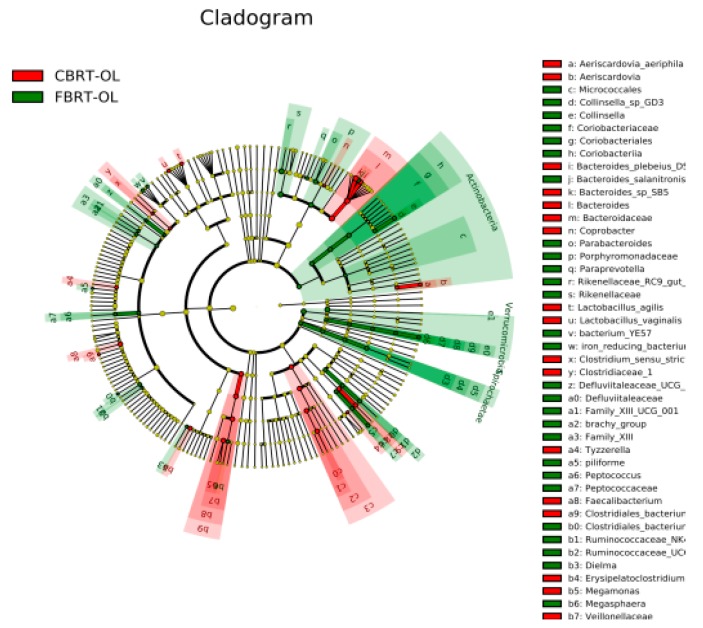
The evolutionary branch tree diagram of significantly different gut microbiota between the CBRT-OL (lignocellulose-low) and FBRT-OL groups.

**Figure 11 animals-10-00107-f011:**
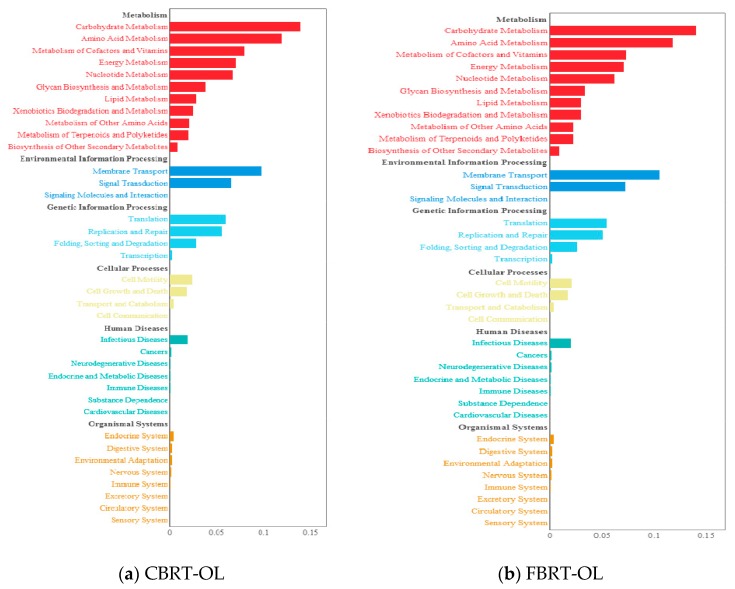
Functional prediction of OTUs of FBRT-OL and CBRT-OL.

**Figure 12 animals-10-00107-f012:**
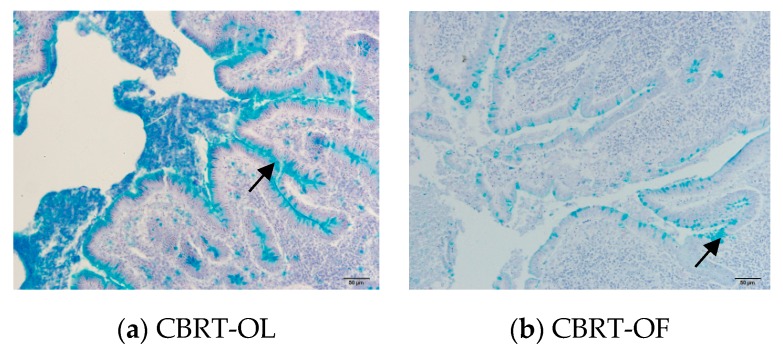
The thickness of the cecum inner mucus layer.

**Figure 13 animals-10-00107-f013:**
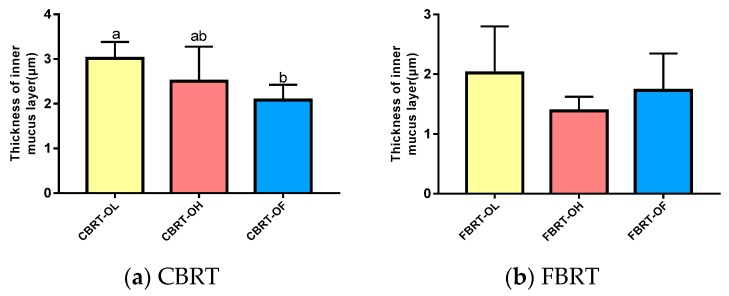
Histograms of the thickness of the cecum inner mucus layer in the different groups.

**Figure 14 animals-10-00107-f014:**
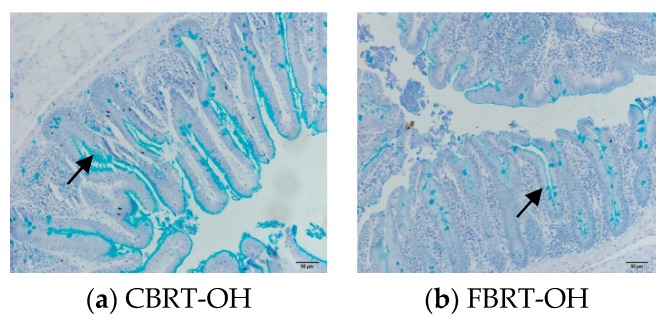
The thickness of the cecum inner mucus layer.

**Figure 15 animals-10-00107-f015:**
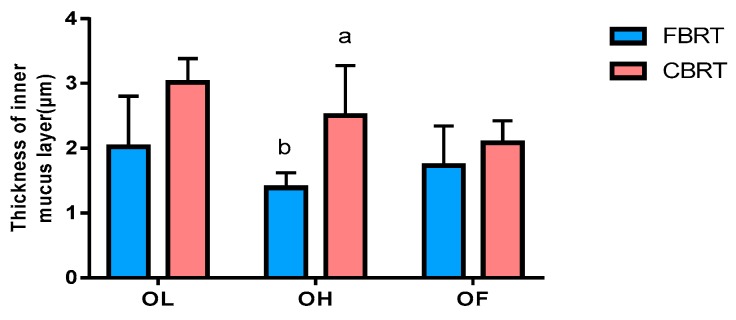
Histograms of the thickness of the cecum inner mucus layer in CBRT and FBRT. a and b represent a significant difference in the groups.

**Table 1 animals-10-00107-t001:** Ingredients and composition of feed used during weeks 1–8.

Ingredients %		Feed Composition %	
Corn	61.95	ME (MJ/kg) ^1^	12.43
Soybean meal	23.7	Crude protein	19.49
Bran	4	Crude fiber	3.21
Soybean oil	1.1	Crude fat	4.27
Corn protein meal	4	Crude ash	5.83
Stone power	1.8	Ca	1.05
CaHPO4	1.3	Total P	0.57
NaCI	0.03	NaCI	0.3
Methionine	0.2	Methionine	0.49
Lysine	0.46	Lysine	0.99
Threonine	0.09	Threonine	0.68
Multivitamin	0.4	Linoleic acid	1.99
Minerals	0.55		
Choline chloride Complex enzyme	0.1 0.05		

^1^ ME: metabolic energy.

**Table 2 animals-10-00107-t002:** Ingredients and feed composition of feed used during weeks 9–20.

		9–15 Weeks			16–20 Weeks	
Ingredients %	OL	OH	OF	OL	OH	OF
Corn	66.28	67.55	65	65.27	66.55	64
Soybean meal	21.58	23.27	20	20.59	22.17	19
Bran	5.14	0.28	10	7.14	2.28	12
Eubiotic lignocellulose^ 1^	2	4	0	2	4	0
Premix ^2^	5	5	5	5	5	5
Calculated analysis (%) ^3^						
ME (MJ/kg)	11.50	11.50	11.50	11.40	11.40	11.40
Crude protein	15.57	15.57	15.57	15.37	15.37	15.37
Crude fiber	4.15	5.02	3.30	4.25	5.11	3.40
Crude fat	3.15	3.05	3.24	3.17	3.07	3.26
Crude ash	2.45	2.35	2.56	2.47	2.37	2.58
Ca	0.99	0.99	0.98	0.98	0.99	0.98
Total P	0.46	0.43	0.49	0.47	0.44	0.50

^1^ The nutritional matrices are as follows. Energy: ~0%; nutrients: (moisture, 8%; crude protein, 0.9%; total dietary fiber (TDF), 88%; crude ash, 1.0%; crude fat, 0.8%; and minerals and trace elements, 1.3%); fiber fractions: (crude fiber, 59%; soluble TDF, 1.3%; NDF, 78%; ADF, 64%; lignin, 25–30%); minerals and trace elements: (Na, 0.03 g/kg; K, 0.5 g/kg; Mg, 0.2 g/kg; Ca, 0.9 g/kg; P, 0.1 g/kg; Cu, 1.2 mg/kg; Fe, 90 mg/kg; Zn, 12 mg/kg; and Mn, 89 mg/kg). ^2^ Premix provided the following (per kg of diet) VA 130~190 KIU; VD3 30~95 KIU; VE ≥ 350 IU; VK ≥ 60 mg; VB1 ≥ 24 mg; VB2 ≥ 100 mg; VB6 ≥ 60 mg; VB12 ≥ 200 mg; nicotinamide ≥ 50 mg; pantothenic acid ≥ 200 mg; folic acid ≥ 12 mg; biotin ≥ 2 mg; choline chloride ≥ 7 mg; Fe 1300~7400 mg; Cu 120~650 mg; Mn 1450~2900 mg; Zn 1250~2900 mg; I 7~95 mg; selenium 6%~9.5%; Ca 12–25%; P (adding phytase) ≥ 2.0%; NaCI 4–10%; methionine ≥ 1.8%; moisture ≤ 10%. ^3^ The composition was calculated but not measured for the feed used.

**Table 3 animals-10-00107-t003:** The additive volume and concentrations of volatile fatty acid standards added to the standard stored solution.

	Acetate	Propionate	Butyrate	Isobutyrate	Isovalerate	Valerate
Additive volume μL	60	40	20	5	5	5
Concentration ^1^ g/L	0.63	0.40	0.19	0.048	0.047	0.047
Mol concentration ^2^ mmol/L	10.41	5.35	2.19	0.54	0.46	0.46

^1^ Concentration of additive standards (g/L) = density of standards (g/mL) × additive volume (μL) ÷ 100. ^2^ Mol concentration (mmol/L) = concentration of additive standards (g/L) ÷ molar mass of standards (g/mol) × 1000.

**Table 4 animals-10-00107-t004:** Comparison of the α-diversity and β-diversity indexes among three groups of CBRT with different added levels of the lignocellulose.

	CBRT-OL	CBRT-OH	CBRT-OF
α-diversity			
ACE	1371.34 ± 97.82 ^B^	1810.22 ± 109.35 ^A^	1631.94 ± 55.75 ^A^
Chao1	1375.74 ± 107.08 ^B^	1873.99 ± 100.77 ^A^	1721.33 ± 95.58 ^A^
Shannon	5.86 ± 0.27 ^B^	6.55 ± 0.067 ^A^	6.57 ± 0.13 ^A^
Simpson	0.92 ± 0.021 ^B^	0.97 ± 0.00094 ^A^	0.97 ± 0.0016 ^A^
β-diversity	0.16 ± 0.032	0.17 ± 0.039	0.17 ± 0.033

Different superscript letters A and B represent an extremely significant difference in the same line.

**Table 5 animals-10-00107-t005:** The comparison of α-diversity and β-diversity among three groups added different levels of the lignocellulose of FBRT.

	FBRT-OL	FBRT-OH	FBRT-OF
α-diversity			
ACE	1773.31 ± 43.86 ^a^	1772.66 ± 58.93 ^a^	1524.77 ± 6.28 ^b^
Chao1	1818.33 ± 51.32	1799.93 ± 76.45	1616.26 ± 30.42
Shannon	7.19 ± 0.056 ^a^	6.87 ± 0.12 ^a,b^	6.66 ± 0.052 ^b^
Simpson	0.98 ± 0.00041	0.97 ± 0.0035	0.97 ± 0.0037
β-diversity	0.15 ± 0.025 ^A,B^	0.23 ± 0.061 ^A^	0.097 ± 0.0043 ^B^

Different superscript letters (A and B) and (a and b) separately represent an extremely significant difference and a significant difference, respectively, in the same line.

**Table 6 animals-10-00107-t006:** The pair-wise comparison of α-diversity and β-diversity between groups of CBRT and FBRT that had the same level of lignocellulose added.

	FBRT-OL	CBRT-OL	FBRT-OH	CBRT-OH	FBRT-OF	CBRT-OF
α-diversity						
ACE	1773.31 ± 43.86 ^A^	1371.34 ± 97.82 ^B^	1772.66 ± 58.93	1810.22± 109.35	1524.77± 6.28	1631.94 ± 55.75
Chao1	1818.33 ± 51.32 ^A^	1375.74 ± 107.08 ^B^	1799.93 ± 76.45	1873.99 ± 100.77	1616.26 ± 30.42	1721.33 ± 95.58
Shannon	7.19 ± 0.056 ^A^	5.86 ± 0.27 ^B^	6.87 ± 0.12	6.55 ± 0.067	6.66 ± 0.052	6.57 ± 0.13
Simpson	0.98 ± 0.00041 ^A^	0.92 ± 0.021 ^B^	0.97 ± 0.0035	0.97 ± 0.00094	0.97 ± 0.0037	0.97 ± 0.0016
β-diversity	0.15 ± 0.025	0.16 ± 0.032	0.23 ± 0.061	0.17 ± 0.039	0.097 ± 0.0043	0.17 ± 0.033

Different superscript letters A and B represent an extremely significant difference in the same line.

**Table 7 animals-10-00107-t007:** The comparison of the development of the small intestine and cecum of FBRT and CBRT.

		OL	OH	OF
FBRT	small intestines (cm)	125.75 ± 5.24 ^a^	120.2 ± 6.69	124.83 ± 4.90
CBRT	small intestines (cm)	108.58 ± 3.91 ^b^	111.3 ± 3.05	112.47 ± 6.41
FBRT	cecum (cm)	13.6 ± 0.50	13.1 ± 0.61	14.92 ± 0.81
CBRT	cecum (cm)	13 ± 1.08	13.67 ± 0.48	12.62 ± 0.34
FBRT	cecum (g)	5.24 ± 0.23	4.38 ± 0.23	5.59 ± 0.26 ^a^
CBRT	cecum (g)	4.47 ± 0.53	4.96 ± 0.36	4.64 ± 0.27 ^b^

Different superscript letters a and b represent a significant difference in the same column.

**Table 8 animals-10-00107-t008:** Effects of added dietary fiber and rearing system on short chain fatty acids (SCFAs) (mmol/L).

	Acetate	Propionate	Butyrate	Isobutyrate	Isovalerate	Valerate
Adding dietary fiber						
CBRT-OL	4.28	1.30	0.64	0.12 ^a,b^	0.13	0.15
CBRT-OH	4.09	1.50	0.58	0.086 ^a^	0.09	0.12
CBRT-OF	5.20	1.82	0.80	0.23 ^b^	0.32	0.17
Pooled SEM	0.54	0.16	0.12	0.026	0.052	0.018
FBRT-OL	4.50	1.58	1.00	0.18	0.27	0.18
FBRT-OH	4.57	1.37	0.63	0.13	0.17	0.14
FBRT-OF	5.67	1.97	0.92	0.14	0.22	0.16
Pooled SEM	0.33	0.16	0.14	0.017	0.026	0.015
Rearing system						
CBRT-OL	4.28	1.30	0.64	0.12	0.13	0.15
FBRT-OL	4.50	1.58	1.00	0.18	0.27	0.18
Pooled SEM	1.14	0.43	0.37	0.046	0.06	0.053
CBRT-OH	4.09	1.50	0.58	0.086 ^b^	0.09	0.12
FBRT-OH	4.57	1.37	0.63	0.13 ^a^	0.17	0.14
Pooled SEM	0.22	0.13	0.10	0.008	0.02	0.008
CBRT-OF	5.20	1.82	0.80	0.23	0.32	0.17
FBRT-OF	4.36	1.46	0.51	0.15	0.18	0.13
Pooled SEM	1.22	0.19	0.21	0.062	0.11	0.036
Added dietary fiber	0.345	0.299	0.678	0.133	0.124	0.498
Rearing system	0.572	0.827	0.264	0.892	0.454	0.272
Added dietary fiber × Rearing system	0.986	0.617	0.438	0.116	0.191	0.455

Different superscript letters a and b represent a significant difference in the same column.

**Table 9 animals-10-00107-t009:** Effects of added dietary fiber and rearing system on the thickness of the cecum inner mucus layer.

	Inner Mucus Layer Thickness (μm)
Added dietary fiber	
CBRT-OL	3.01 ^a^
CBRT-OH	2.50 ^a,b^
CBRT-OF	2.09 ^b^
Pooled SEM	0.18
FBRT-OL	2.02
FBRT-OH	1.39
FBRT-OF	1.73
Pooled SEM	0.17
Rearing system	
CBRT-OL	3.01
FBRT-OL	2.02
Pooled SEM	0.28
CBRT-OH	2.50 ^a^
FBRT-OH	1.39 ^b^
Pooled SEM	0.23
CBRT-OF	2.09
FBRT-OF	1.73
Pooled SEM	0.22
Added dietary fiber	0.032
Rearing systems	0.003
Added dietary fiber × Rearing system	0.46

Different superscript letters a and b represent a significant difference in the same column.

**Table 10 animals-10-00107-t010:** Effects of added dietary fiber and rearing system on glucagon like peptide-1 (GLP-1).

	GLP-1 (pmol/L)
Added dietary fiber	
CBRT-OL	12.46
CBRT-OH	11.58
CBRT-OF	12.05
Pooled SEM	0.25
FBRT-OL	8.94
FBRT-OH	9.91
FBRT-OF	10.20
Pooled SEM	0.23
Rearing system	
CBRT-OL	12.46 ^A^
FBRT-OL	8.94 ^B^
Pooled SEM	0.15
CBRT-OH	11.58 ^A^
FBRT-OH	9.91 ^B^
Pooled SEM	0.32
CBRT-OF	12.05 ^A^
FBRT-OF	10.20 ^B^
Pooled SEM	0.32
Added dietary fiber	0.46
Rearing system	0.00
Added dietary fiber × Rearing system	0.051

Different superscript letters (A and B) represent an extremely significant difference in the same column.

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
