# Peer review of "Effects of Added Dietary Fiber and Rearing System on the Gut Microbial Diversity and Gut Health of Chickens"

_animals, 2020, doi:10.3390/ani10010107_

Round 1

Reviewer 1 Report

Except for a few minor editorial issues this is a well written manuscript that could be of major interest to the worldwide poultry industry because of the ability of the product to increase nutritive values to free-range chickens; thus improving animal welfare.  However, there is one major issue that I have with this manuscript and that is that the experiment was conducted once and not repeated.  Now expect the time is an issue with repeating the experiment because this one trial was months long, but for me to believe that the improvements in microbial diversity and overall gut health I must see that these improvements are consistent between flocks of birds.  These data show wondaful results for the present flock, but are the 'trends' consistent?  Without repeating the experiment, these data are biologically interesting, but I want to see biologically significance  

A few minor editorial issues that require attention:

1.  Line 76: Change to The Bian chicken is an endogenous Chinese line of poultry...

2.  Line 97: Change to 108 one-day old Bian male chickens..

3.  Line 202: Provide the source of the chicken GLP-1 kit. 

Author Response

Dear teacher,

Thank you very much. Sorry, I spent so many days in revising the manuscript to delay your review . I have uploaded the revised version and my responses to you and please check it. Thank you again .

Kind regards,

Linyue Hou

Reviewer 2 Report

The content of the paper is interesting in my opinion. It shows the effect of dietary fiber on microbiome diversification in native Chinese broilers.

L:88 - China began to import it in 2012.- remove it is useless information

L90-93 – please be more specific and rewrite aim of the study

L101 – in total all nutrient = around 62% what else was present? Please check the OptiCell composition it seems that there is mistake there.

L107-120: treatments and how birds were assigned to them, what was body weight at the beginning of the trial should be described in details

L122: provide diet composition and nutritional value for diet used from 1-8 week

All tables – improve the title of all tables they have to be more informative

Table 1: provide a determined nutrient composition, remove “%” and put it in first row. Provide premix composition.

L:139 what does mean “chickens were humanely slaughtered”

Table 2: I don’t see rationality for this table

L222-231; please provided appropriate references for R-Cran and used packages. Additionally in my opinion it should be 3x2 study. You have OptiCell (three levels 0, L and H) and “feeding modes” (grassland vs. cage)

L232: you should start with the performance of birds – BWG, FCR, FI

Table 7; provide units for SCFA, remove SD provide pooled SEM

I don’t understand what is a difference between Tab. 7 and 8 and 9 vs. 10

The discussion should be rewritten, authors should discus results keeping in mind that it is a 2x3 study design. You should focus on interactions and if they will be insignificant you should discuss the main effects only.

Author Response

Dear teacher,

Thank you very much. You comments are quite right and very useful for me. I have revised  the manuscript refer to thes comments. I have uploaded the revised version and response to you . Thank you again for everything you have done.

Kind regards

Linyue Hou

Round 2

Reviewer 1 Report

The author's argument for the 'trends' is exactly the pint I am making about repeating an experiment more than one time.  Microbial diversity does change in relation to diet, age, breed, etc. but running an experiment one time and then making a claim that a product is 'good' for performance tells us nothing.  The experiment must be repeated with a different flock of birds (same breed, age) to see if Opticell induces the same trend.  The results do NOT have to be identical, but the trend to better performance and gut health is what is important.  Presenting the data from both experiments with prove your point.  AS it stands, you do not have convincing data from one experiment

Author Response

Dear teacher,

Thank you very much. I have revised the manuscript according to your comments and suggestions. I have uploaded the revised file and a cover letter to you, and I request that you can check it in your spare time. Thank you again.

The author's argument for the 'trends' is exactly the pint I am making about repeating an experiment more than one time. Microbial diversity does change in relation to diet, age, breed, etc. but running an experiment one time and then making a claim that a product is 'good' for performance tells us nothing. The experiment must be repeated with a different flock of birds (same breed, age) to see if Opticell induces the same trend. The results do NOT have to be identical, but the trend to better performance and gut health is what is important. Presenting the data from both experiments with prove your point.  AS it stands, you do not have convincing data from one experiment

Response: Yes, teacher, you are right. I slightly mistook your meaning last time and I think I get it this time. In my opinion, just as a large number of clinical trials are needed before the effective role or side effect of a drug is finally identified, and then it can be put into production. Similarly, we can’t definitely claim that a product is 'good' for gut microbial diversity and gut health only repeat an experiment unless Opticell induces the same trend when repeated with a different flock of birds, because the improvement of microbial diversity and gut health maybe resulted from other factors such as plant fiber in wild besides Opticell.

I have revised the title, abstract and conclusion. For example, I have deleted the sentence L48 “an appropriate amount of OptiCell benefitted the microbial diversity and health of chickens” in abstract. In addition, I have replaced “can” with “maybe” in some places of revised file. Please see L4-6, L25-27, L42-44 , L437, L471 and L566-567. But I don’t know whether these revisions are suitable or not and I sincerely hope you point out mistakes and give some suggestions please. Thank you very much.

Best wishes,

Linyue Hou

Reviewer 2 Report

Dear Authors,

I think that your manuscript new needs English editing, I’m not a native speaker but I see a lot of typographical, grammar errors. In many places, I see a problem with professional language, e.g. “feeding modes”, bean oil, stone meal, compound enzyme, positive correlation, endogenous chickens, etc.

You provided an answer for most of my remarks but still, I don’t see the following information in this version of manuscript:

Point 3: L101 – in total all nutrient = around 62% what else was present? Please check the OptiCell composition it seems that there is a mistake there.

Point 4: L107-120: treatments and how birds were assigned to them, what was body weight at the beginning of the trial should be described in detail.

When I was asking about that I thought about this moment “At the beginning of the ninth week, the chickens were regrouped. Some of the chickens were 121 transferred into three-step cages to continue being caged with …”

Point 5:L122: provide diet composition and nutritional value for diet used from 1-8 week

Please improve tables with nutrients composition – the energy it is not a correct term, please provide determined nutrient composition instead of calculated.

Point 12:Table 7; provide units for SCFA, remove SD provide pooled SEM

Response12 : OK, teacher. I have provided units for SCFA, pooled SEM, main effects and interaction in Table 8. Please see L432 (Table 8), L452 (Table 9) and L483 (Table 10) .

And this is how all table should be presented

Response13: OK, teacher. Original Tab.7 and 9 were the comparison among groups of CBRT or FBRT under the same feeding mode. Original Tab. 8 and 10 wre the comparison between the groups of CBRT and FBRT under the same adding OptiCell level. They have been rebuilt. Please see L432(Table 8) and L452(Table 9) .

But I don’t understand them, it is very difficult to follow these tables, organization is very awkward

Author Response

Dear teacher,

Thank you very much! I have carefully revised the manuscript according to the comments and suggestions of Academic Editor  and reviewers. I have uoloaded the revised file and cover letter, and I request you to check it in your spare time please. Thank you again.

I think that your manuscript new needs English editing, I’m not a native speaker but I see a lot of typographical, grammar errors. In many places, I see a problem with professional language, e.g. “feeding modes”, bean oil, stone meal, compound enzyme, positive correlation, endogenous chickens, etc.

Response 1: Sorry, teacher, there were a lot of grammar errors and I need more skills in high quality English writing. My manuscript has been checked use a professional English editing software. In addition, the problem with professional language has been addressed. For example, I have changed all “feeding modes” to “rearing systems”, changed “bean oil” to “soybean oil”, changed “stone meal” to “stone powder”, changed “compound enzyme” to “complex enzyme” and changed “endogenous chickens” to “a Chinese native breed”. Please see L141-143 (Table 1) and L85.

2.You provided an answer for most of my remarks but still, I don’t see the following information in this version of manuscript:

Point 3: L101 – in total all nutrient = around 62% what else was present? Please check the OptiCell composition it seems that there is a mistake there.

Response 2: Sorry, teacher. The nutrients composition of OptiCell were provided by Beijing e-feed & e-vet cooperation. Please see L146-147 (under the Table 2).

3.Point 4: L107-120: treatments and how birds were assigned to them, what was body weight at the beginning of the trial should be described in detail.

When I was asking about that I thought about this moment “At the beginning of the ninth week, the chickens were regrouped. Some of the chickens were transferred into three-step cages to continue being caged with …”

Response 3: Yes, teacher. I have described them in detail.

4.Point 5:L122: provide diet composition and nutritional value for diet used from 1-8 week

Please improve tables with nutrients composition – the energy it is not a correct term, please provide determined nutrient composition instead of calculated.

Response 4: OK, teacher. The ”energy” should be “metabolic energy(ME)”. In addition, we bought the feed used during the 1-8 weeks from Shiyang Feed Ltd (Taigu, Shanxi) and the nutrients composition of feed was provided by company. We bought OptiCell (Beijing), corn, soybean meal, bran and premix (Shiyang Feed Ltd) to make three kinds of feed with equal ME and protein, but different amounts of fiber during 9-20 weeks. Some nutrient composition of these raw materials were determined and provided by companies. However, we didn’t determine the ME of the mix feed because we lacked machine. So we first calculated the ratio of raw materials of three feed at the same ME and protein level set according to NRC, and then calculated nutrients composition of the mix feed. It should be some differences between the theoretical values and determined results. You are right, teacher, so we have marked “Calculated analysis (%)” in table2.

5.The nutrient composition and nutritional value for diet used from5.Point 12:Table 7; provide units for SCFA, remove SD provide pooled SEM

Response12 : OK, teacher. I have provided units for SCFA, pooled SEM, main effects and interaction in Table 8. Please see L432 (Table 8), L452 (Table 9) and L483 (Table 10) .

And this is how all table should be presented.

Response 5: Thank you, teacher. I get it. This is very useful for me and I believe I can  successfully address it if I meet this type of experiment design again.

6. Response13: OK, teacher. Original Tab.7 and 9 were the comparison among groups of CBRT or FBRT under the same feeding mode. Original Tab. 8 and 10 were the comparison between the groups of CBRT and FBRT under the same adding OptiCell level. They have been rebuilt. Please see L432(Table 8) and L452(Table 9) .

But I don’t understand them, it is very difficult to follow these tables, organization is very awkward

Response 6: Sorry, teacher, I didn't make it clear last time. The previous Tab.7 & Tab 8 have been replaced by the new Tab.8 and the previous Tab.9 & Tab 10 have been replaced by the new Tab.9 now.

Kind regards,

Linyue Hou 
